# Digital technology and patient and public involvement (PPI) in routine care and clinical research—A pilot study

Yang Chen[1,2]☯*, Ali A. Hosin[3]☯, Marc J. George[3], Folkert W. Asselbergs[1,2,4,5], Anoop D. Shah[1,2,3]

1 Institute of Health Informatics, Faculty of Population Health Sciences, University College London, London, United Kingdom, 2 Clinical Research Informatics Unit, University College London Hospitals, London, United Kingdom, 3 Clinical Pharmacology Department, University College London Hospitals, London, United Kingdom, 4 Division Heart & Lungs, Department of Cardiology, University Medical Center Utrecht, Utrecht University, Utrecht, The Netherlands, 5 Institute of Cardiovascular Science, Faculty of Population Health Sciences, University College London, London, United Kingdom

☯ These authors contributed equally to this work.
* yang.a.chen@ucl.ac.uk

**Data Availability Statement:** All relevant data are within the paper and its Supporting Information files.

## Abstract

### Background

Patient and public involvement (PPI) has growing impact on the design of clinical care and research studies. There remains underreporting of formal PPI events including views related to using digital tools. This study aimed to assess the feasibility of hosting a hybrid PPI event to gather views on the use of digital tools in clinical care and research.

### Methods

A PPI focus day was held following local procedures and published recommendations related to advertisement, communication and delivery. Two exemplar projects were used as the basis for discussions and qualitative and quantitative data was collected.

### Results

32 individuals expressed interest in the PPI day and 9 were selected to attend. 3 participated in person and 6 via an online video-calling platform. Selected written and verbal feedback was collected on two digitally themed projects and on the event itself. The overall quality and interactivity for the event was rated as 4/5 for those who attended in person and 4.5/5 and 4.8/5 respectively, for those who attended remotely.

### Conclusions

A hybrid PPI event is feasible and offers a flexible format to capture the views of patients. The overall enthusiasm for digital tools amongst patients in routine care and clinical research is high, though further work and standardised, systematic reporting of PPI events is required.

**Funding:** This study was supported by a UCLH Biomedical Research Centre, Grant Number: BRC849/PPI/AH/104990, awarded to AH.

**Competing interests:** The authors have declared that no competing interests exist.

## Introduction

Patient and public involvement (PPI) work has increasingly gained prominence in healthcare settings. Organisations such as the National Institute for Health Research (NIHR) [1] and the Patient Centered Outcomes Research Institute (PCORI) [2] advocate for PPI and engagement (PPIE) to play a central role in research design. Greater PPI participation in routine clinical care was also highlighted in a recent scoping review [3]. Developing alongside these innovations has been the increasing use of digital technologies in healthcare [4]. Most recently, a systematic review of patient and public views on artificial intelligence (AI)–at the frontier of digital technologies–highlighted that whilst there were positive attitudes toward AI, reservations remained and human supervision of AI in healthcare decision-making was a preferred [5]. As PPI and digital tools continue to increase their mark in healthcare, there is growing need for research that lies at the intersection of these two trends. This study helps to address this gap in the literature and had the following aims:

1. To determine the feasibility of hosting a PPI event using digital tools and to gather feedback on the usefulness of a hybrid participation format

2. To gather patient views on the role of digital technologies in routine care delivery and clinical research, with specific focus on remote monitoring of blood pressure, consent processes and patient reported outcome measures (PROMs) as exemplar themes.

## Methods

To minimise digital exclusion [6], patients were recruited through online and offline means (UCLH NIHR Biomedical Research Centre, British Heart Foundation Networks and UCLH clinical pharmacology clinic). Box 1 outlines the specific projects discussed during the event.

> ## Box 1. Projects discussed during the PPI event
>
> Our discussions were framed around two projects that were both in their early design/ pilot phase:
>
> 1. A quality improvement project (QIP) which related to the remote monitoring of blood pressure through a bespoke smartphone application or website and safe data linkage to the patient electronic care record. The specific platform discussed which would be used to host remotely collected data was the 'Mycare UCLH patient portal' [7].
>
> 2. A research study proposal examining the use of digital tools during hospital stays to both as interventions such as clinician-facing alerts, as well as platforms to document patient views.

After registration of interest, individuals were invited to complete a short survey to confirm their availability and to optionally volunteer anonymised demographic data to aid the selection of a diverse cohort of PPI participants. Other background information including social determinants of health were not collected in order to prioritise accessibility and ease of completion. The venue was chosen to ensure adherence to social distancing with appropriate audio-visual facilities to allow two-way communication between live and remote participants. An upper limit of 10 participants for the event was used based on previous guidance for focus group size

[8]. In line with NHS governance policy, Microsoft Teams was the chosen online platform [9]. A shortlist of 10 participants was purposively selected to generate a balanced group. This was based on protected characteristics such as age and gender, as well as pre-event confidence in using digital tools.

A scribe was appointed amongst the study group who contemporaneously recorded the key themes of the verbal discussion. Direct quotations were used, and quantitative feedback was collected in selected data capture forms that were distributed. The pre-attendance short survey allowed prospective participants to provide written consent before the event, and verbal consent was taken on the day for use of non-attributable quotations that arose from discussions. A short feedback form was distributed to participants after the event. The delivery and contents of the day including slide sets used to trigger co-design discussions are available in the S1 File. This study is reported according to the GRIPP2 long form where relevant and a checklist is supplied in the S1 File [10]. In accordance with the UK Health Research Authority decision tool on ethical requirements for research [11] and specific guidance from the National Institute for Health and Care Research on the ethics of PPI events, no approval from an ethics committee was sought [12].

## Results

A total of 32 participants expressed interest in the event. 19 patients were available to attend on the day and a summary of their characteristics is found in Table 1. A total of 9 participants attended on the day (3 face-to-face, 6 via Microsoft Teams).

Outcomes generated from the day and in feedback were grouped into qualitative and quantitative domains.

### Qualitative outcomes

The following selected quotations were informative in driving changes to the proposed care pathway for remote monitoring of BP:

"*You can spot trends, I wore a heart monitor, especially post-surgery*

*But the downsides are—how do you motivate people, is it anxiety provoking*"—*Participant 3*

"*I'm really interested in what stress and anxiety can do to your body but what about doing it everyday obsessively and a bit more than I need to. At what point should I be making contact with my GP.*"—*Participant 8*

**Table 1. Baseline characteristics of study participants.**

|  | Available participants (n = 19) | Actual participants (n = 9) |
|---|---|---|
| Age (SD) | 62 (13.1) | 60.5 (15.8) |
| Gender | Male n = 10, (52.6%) | Male n = 5 (55.6%) |
|  | Female n = 9 (47.4%) | Female n = 4 (44.4%) |
| Ethnicity | 89.5% White British or White other (n = 17) | 88.9% White British (n = 8) |
|  | 10.5% Other (Pakistani, North African) (n = 2) | 11.1% Other (n = 1) |
| Confidence with digital tools (out of 5 point Likert scale) (SD) | 4.34 (0.9) | 4 (1) |

These quotes were indicative of the enthusiasm for remote monitoring that the group had and their inquisitiveness about how such their BP data would be shared with their usual care team. However, there was an emphasis additionally placed on safeguarding required to mitigate against over-measurement and the possible anxiety that elevated readings could create, though this was balanced with points made about elevated readings being positively viewed, in terms of being a motivating factor to engage in lifestyle behaviours such as exercise to reduce BP.

As a result of the group discussions, the following changes to the remote BP project were made:

- Adjusting the pathway to share BP data and feedback with more members of the care team

- Refinement of the data entry system to include app-based reminders and advice on when to recommend initiating further discussions

- A proposal to incorporate elements of counselling and support at the time of diagnosis

In terms of the research proposal, discussions focused primarily on the ethics and logistics of delivering a clinician-facing digital alert as an intervention within a research study. A selected quotation with regards to waiver of consent was:

> "*seems the most practical way of avoiding any behavioural effects that come from knowing you are in a research trial*" Participant 8

This quote summarised the consensus view of the group in terms of how to balance the feasibility and safety of doing research studies of electronic alerts aimed at clinicians which seek to change their behaviour and downstream management strategies that can affect patients, In this regard, the group understood that thresholds for tolerating waived consent may vary case by case. There was a majority consensus for waived consent in the context of comparing existing non-pharmaceutical interventions such as fluid restriction versus no restriction, given the patient is the final arbiter of oral fluid intake. Support for waiver was not unanimous though and so additional review and communication strategies were sought by the trial management group because of the diverse and helpful patient feedback.

## Quantitative outcomes

For the research proposal, different models of consent were rated on a 5-point Likert scale for suitability and responses were individually completed following discussion (Table 2). The average suitability for each consent model was: waiver of consent: 4.5/5; research registry: 3/5; verbal consent: 2.7/5. A significant amount of missing data was noted for participants who attended remotely.

Table 3 additionally summarises views collected on patient reported outcome measures (PROMs). Data was only available for the three participants who attended in person.

## Feedback on PPI event and impact of context

The event was held on December 3rd, 2021, and strict Covid-19 protocols were observed for those who attended in person. The hybrid format proved challenging in relation to timings observed for the day. Whilst the majority of participants (6 out of 9) were logged in remotely, ensuring an equitable distribution of time was difficult though partly helped by all participants having webcam access. Table 4 highlights feedback collected from 8 participants after the event with similar mean ratings between in person (overall rating 4/5, interactivity 4/5) and remote attendance (overall quality 4.5/5, interactivity 4.8/5).

**Table 2. Patient views on different models of consent for the research proposal.** Participants (i) to (iii) were those who attended in person. First, participants were asked to rank the models of consent in order of preference (1 = most preferred, 3 = least preferred). Next, participants were asked to rate each model out of 5 for suitability (1 = not at all suitable; 5 = entirely suitable).

| Participant | Rank model of consent (1 = most preferred, 3 = least preferred) | | | Suitability of each model (out of 5, 5 = entirely suitable, 1 = not at all suitable) | | |
|---|---|---|---|---|---|---|
| | Waiver | Research registry | Verbal consent | Waiver | Research registry | Verbal consent |
| **(i)** | 1 | 2 | 3 | 5 | 2 | 1 |
| **(ii)** | 2 | 1 | 3 | 5 | 5 | 5 |
| **(iii)** | 1 | 2 | 3 | 5 | 3 | 2 |
| **(iv)** | 1 | 2 | 3 | 5 | 2 | 1 |
| **(v)** | 1 | 2 | 3 | 5 | 3 | 2 |
| **(vi)** | 3 | 2 | 1 | 2 | 3 | 5 |

**Table 3. Participant views on PROMs.** Participants were asked to what extent they agreed with specific statements on how patient reported outcome measures (PROMs) may be used (1 = not at all, 5 = very much agree).

| Participant | To what extent do you agree with the following (out of 5, 5 = very much agree, 1 = not at all) | |
|---|---|---|
| | I wish to be part of the data documentation process with regards to fluid balance and quality of life | I wish to take part in future research where there is full data completeness and therefore support quality improvement projects that improve routine care to ensure this can happen |
| **(i)** | 5 | 5 |
| **(ii)** | 1 | 3 |
| **(iii)** | 5 | 5 |

**Table 4. Summary of rated quality and interactivity of the day from participants.**

| Participant | Each question asked on a five point Likert scale where 5 was most positive and 1 was least | |
|---|---|---|
| | Overall rating of the day (average of morning and afternoon session) | Level of interactivity (average of morning and afternoon session) |
| **(i)** | 5 | 5 |
| **(ii)** | 4 | 4 |
| **(iii)** | 3 | 3 |
| **(iv)** | 4 | 5 |
| **(v)** | 5 | 5 |
| **(vi)** | 4 | 4 |
| **(vii)** | 5 | 5 |

## Discussion

We have shown in this study that a hybrid PPI event was feasible to deliver and received positive feedback that was similar between in person and remote attendance. The engagement from participants and their views was used to shape two distinct projects in their design phase.

## Comparison with literature

We are unaware of existing published reports of hybrid PPI events which not only examine digital tools as a theme, but which used such tools to allow the safe conduct of the day. Whilst reducing commute times and accommodating those who are risk averse in the pandemic era are advantages, there is the caveat of ensuring factors such as digital exclusion [7] do not reinforce biases and underrepresentation of key groups.

Thus systematic design and delivery of PPI will require dedicated focus to equality, diversity and inclusion, particularly as upstream PPI begins to shape both research (including improving recruitment to clinical trials [13]) and routine service design (such as increased participation in co-design).

When compared to the distribution of current PPI events, [14] it is unclear if participation levels are representative of patients affected by each condition or treatment studied, or if the distribution of known and published PPI events match up to patient priorities more generally.

## Limitations

Our findings should be considered in light of several limitations. First, our sample size was small and there is the risk of ascertainment bias, given that recruitment of patients occurred through a limited number of sources. We mitigated this in part through a pre-event survey but nevertheless the views of the focus day may not be representative of the wider population, something that all PPI is at risk of [15].

Second, our event was time limited–at 4.5hours in duration including breaks, we sought to balance factors such as breadth and depth of coverage as well as risk of fatigue. Written feedback collected for the day noted that timing was tight and the afternoon session in particularly was abbreviated to adhere to time, ensuring that for in-person attendees would not miss their return travel home.

Third, the data collection form used to quantify views was a 1–5 Likert scale. We elected to forgo using a formal validated measuring instrument in favour of ease and speed of use. As a pilot event, we regarded the data collection exercise as more exploratory in nature.

Fourth, we did not use a formal framework to support our PPI day design. This was motivated by pragmatism, with existing evidence to suggest that most recognised frameworks have poor useability [16]. However, there remain questions regarding how best to harness PPI for implementation-focused clinical and research questions and previous work in the field has offered suggestions which require further exploration [17].

## Future directions

The optimal way to harness PPI to help implement digital technologies and to inform how we test them as part of pragmatic research, [18] remain areas for further work. PPI efforts will be central to the maintenance of trust in their use. [19, 20].

## Conclusion

A hybrid PPI event is feasible and offers a flexible format to capture patient and public views when planning clinical service design, quality improvement or research projects. It may be of particular benefit in capturing the views of those unable to attend otherwise, for reasons including mobility and clinical vulnerability. Striking a balance of 'informal PPI' with pragmatic recruitment and delivery of more formal events will be a consideration for all research and clinical teams. The challenges related to adequate time to facilitate discussions and to capture views during future hybrid formats will require additional staff training and sharing of feedback and experiences, allied to better reporting of PPI events.

## Supporting information

**S1 File. Additional material including information on study delivery, slides used and GRIPP2 reporting checklist.**
(DOCX)

## Acknowledgments

We are grateful to the British Heart Foundation Heart Voices network were instrumental in helping to recruit patients and the assistance from Chloe Goldman in this regard. We are additionally grateful to Tom Lumbers for help with reviewing slides for the PPI day and contributing to the content and themes for the research element of the day. Lastly, we are grateful to all patients who expressed interest in the day and who kindly volunteered their time and views in helping to support this study and to inform the design of the two projects that have been discussed during this paper.

## Author Contributions

**Conceptualization:** Yang Chen, Ali A. Hosin.

**Data curation:** Yang Chen, Ali A. Hosin.

**Formal analysis:** Ali A. Hosin.

**Funding acquisition:** Ali A. Hosin, Marc J. George.

**Methodology:** Yang Chen, Marc J. George, Anoop D. Shah.

**Project administration:** Ali A. Hosin, Marc J. George, Anoop D. Shah.

**Resources:** Yang Chen, Ali A. Hosin, Folkert W. Asselbergs.

**Supervision:** Folkert W. Asselbergs, Anoop D. Shah.

**Writing – original draft:** Yang Chen, Ali A. Hosin.

**Writing – review & editing:** Yang Chen, Ali A. Hosin, Marc J. George, Folkert W. Asselbergs, Anoop D. Shah.

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
