## [Decision Letter · Decision Letter 0]

17 Oct 2022

PONE-D-22-13809Digital technology and patient and public involvement (PPI) in routine care and clinical research - a pilot study

PLOS ONE

Dear Dr. Chen,

Thank you for submitting your manuscript to PLOS ONE. After careful consideration, we feel that it has merit but does not fully meet PLOS ONE’s publication criteria as it currently stands. Therefore, we invite you to submit a revised version of the manuscript that addresses the points raised during the review process.

We look forward to receiving your revised manuscript.

Kind regards,

Giuseppe Limongelli

Academic Editor

PLOS ONE

Journal Requirements:

a) Did participants provide their written or verbal informed consent to participate in this study?

b) If consent was verbal, please explain i) why written consent was not obtained, ii) how you documented participant consent, and iii) whether the ethics committees/IRB approved this consent procedure."

"ADS is supported by a postdoctoral fellowship from THIS Institute, grants from NIHR and the UCL British Heart Foundation Research Accelerator. FWA and YC are supported by UCL Hospitals NIHR Biomedical Research Centre. AH is supported by a Health Education Eng-land Topol Digital Fellowship. The PPI event was kindly supported by funds from the NIHR UCLH BRC PPI bursary fund. The British Heart Foundation Heart Voices network were in-strumental in helping to recruit patients and we are grateful for the assistance of Chloe Gold-man in this regard. We are additionally grateful to Tom Lumbers for help with reviewing slides for the PPI day and contributing to the content and themes for the research element of the day. Lastly, we are grateful to all patients who expressed interest in the day and who kindly volunteered their time and views in helping to support this study and to inform the design of the two projects that have been discussed during this paper."

"The author(s) received no specific funding for this work"

Reviewers' comments:

Reviewer's Responses to Questions

**Comments to the Author**

1. Is the manuscript technically sound, and do the data support the conclusions?

Reviewer #1: Yes

Reviewer #2: Yes

2. Has the statistical analysis been performed appropriately and rigorously? 

Reviewer #1: N/A

Reviewer #2: N/A

3. Have the authors made all data underlying the findings in their manuscript fully available?

Reviewer #1: Yes

Reviewer #2: Yes

4. Is the manuscript presented in an intelligible fashion and written in standard English?

Reviewer #1: Yes

Reviewer #2: Yes

5. Review Comments to the Author

Reviewer #1: This study contributes to the literature about hybrid PPI events to understand patients' points of view. However, it would make a more substantial contribution if a more comprehensive analysis of the qualitative findings were presented, not just presenting informants' quotations without further explanation of what they imply.

Reviewer #2: The manuscript by Yang Chen and Colleagues is a well written and sound exploratory report, and may offer some relevant messages for assessing PPI. Patient and public involvement and engagement is gradually becoming part of standard cardiovascular and cancer research and an increasing number of statements are being released regularly to support it.

However, methods description and results presentation still have room for improvement and may be addressed. Please find my suggestions below.

- Little information is currently present regarding the steps that led to the final study cohort selection. It would be useful to know how the final study cohort was selected and derived (especially in light of the pre-event survey that Researchers conducted before the pilot study).

-Do Authors have more information regarding social variables of participants or do they have access to healthcare data?

-From a practical point of view, it would be useful to have more information regarding study timeline, staff training, and study and interview duration.

6. PLOS authors have the option to publish the peer review history of their article (what does this mean?). If published, this will include your full peer review and any attached files.

Reviewer #1: No

Reviewer #2: No

---

## [Author Response · Author response to Decision Letter 0]

26 Oct 2022

Reviewer #1: This study contributes to the literature about hybrid PPI events to understand patients' points of view. However, it would make a more substantial contribution if a more comprehensive analysis of the qualitative findings were presented, not just presenting informants' quotations without further explanation of what they imply.

Thank you. We have added the following pieces of analysis:

“These quotes were indicative of the enthusiasm for remote monitoring that the group had and their inquisitiveness about how such their BP data would be shared with their usual care team. However, there was an emphasis additionally placed on safeguarding required to mitigate against over-measurement and the possible anxiety that elevated readings could create, though this was balanced with points made about elevated readings being positively viewed, in terms of being a motivating factor to engage in lifestyle behaviours such as exercise to reduce BP.”

“This quote summarised the consensus view of the group in terms of how to balance the feasibility and safety of doing research studies of electronic alerts aimed at clinicians which seek to change their behaviour and downstream management strategies that can affect patients, In this regard, the group understood that thresholds for tolerating waived consent may vary case by case. There was a majority consensus for waived consent in the context of comparing existing non-pharmaceutical interventions such as fluid restriction versus no restriction, given the patient is the final arbiter of oral fluid intake.”

Reviewer #2: The manuscript by Yang Chen and Colleagues is a well written and sound exploratory report, and may offer some relevant messages for assessing PPI. Patient and public involvement and engagement is gradually becoming part of standard cardiovascular and cancer research and an increasing number of statements are being released regularly to support it.

Thank you. 

However, methods description and results presentation still have room for improvement and may be addressed. Please find my suggestions below.

- Little information is currently present regarding the steps that led to the final study cohort selection. It would be useful to know how the final study cohort was selected and derived (especially in light of the pre-event survey that Researchers conducted before the pilot study).

We have added the following:

“A shortlist of 10 participants was purposively selected to generate a balanced group. This was based on protected characteristics such as age and gender, as well as pre-event confidence in using digital tools.”

-Do Authors have more information regarding social variables of participants or do they have access to healthcare data?

We have added the following:

“Other background information including social determinants of health were not collected in order to prioritise accessibility and ease of completion.”

-From a practical point of view, it would be useful to have more information regarding study timeline, staff training, and study and interview duration.

We have added these relevant details to the supplementary material. 

Response to editorial comments

a) Did participants provide their written or verbal informed consent to participate in this study?

b) If consent was verbal, please explain i) why written consent was not obtained, ii) how you documented participant consent, and iii) whether the ethics committees/IRB approved this consent procedure."

We have added the following:

“The pre-attendance short survey allowed prospective participants to provide written consent before the event, and verbal consent was taken on the day for use of non-attributable quotations that arose from discussions.”

“In accordance with the UK Health Research Authority decision tool on ethical requirements for research [11] and specific guidance from the National Institute for Health and Care Research on the ethics of PPI events, no approval from an ethics committee was sought.[12]”

"ADS is supported by a postdoctoral fellowship from THIS Institute, grants from NIHR and the UCL British Heart Foundation Research Accelerator. FWA and YC are supported by UCL Hospitals NIHR Biomedical Research Centre. AH is supported by a Health Education Eng-land Topol Digital Fellowship. The PPI event was kindly supported by funds from the NIHR UCLH BRC PPI bursary fund. The British Heart Foundation Heart Voices network were in-strumental in helping to recruit patients and we are grateful for the assistance of Chloe Gold-man in this regard. We are additionally grateful to Tom Lumbers for help with reviewing slides for the PPI day and contributing to the content and themes for the research element of the day. Lastly, we are grateful to all patients who expressed interest in the day and who kindly volunteered their time and views in helping to support this study and to inform the design of the two projects that have been discussed during this paper."

"The author(s) received no specific funding for this work"

We have removed the relevant sentences related to funding in the acknowledgements section.

In the funding statement, we have updated this to:

This work was directly supported by a BRC PPI Bursary Award - BRC849/PPI/AH/104990

---

## [Decision Letter · Decision Letter 1]

14 Nov 2022

Digital technology and patient and public involvement (PPI) in routine care and clinical research - a pilot study

PONE-D-22-13809R1

Dear Dr. Chen,

We’re pleased to inform you that your manuscript has been judged scientifically suitable for publication and will be formally accepted for publication once it meets all outstanding technical requirements.

Kind regards,

Giuseppe Limongelli

Academic Editor

PLOS ONE

Additional Editor Comments (optional):

Reviewers' comments:

Reviewer's Responses to Questions

**Comments to the Author**

1. If the authors have adequately addressed your comments raised in a previous round of review and you feel that this manuscript is now acceptable for publication, you may indicate that here to bypass the “Comments to the Author” section, enter your conflict of interest statement in the “Confidential to Editor” section, and submit your "Accept" recommendation.

Reviewer #1: All comments have been addressed

Reviewer #2: All comments have been addressed

2. Is the manuscript technically sound, and do the data support the conclusions?

Reviewer #1: Partly

Reviewer #2: Yes

3. Has the statistical analysis been performed appropriately and rigorously? 

Reviewer #1: N/A

Reviewer #2: N/A

4. Have the authors made all data underlying the findings in their manuscript fully available?

Reviewer #1: Yes

Reviewer #2: Yes

5. Is the manuscript presented in an intelligible fashion and written in standard English?

Reviewer #1: Yes

Reviewer #2: Yes

6. Review Comments to the Author

Reviewer #1: The authors have well-addressed the reviewers' responses. However, the additional explanation to the quotation would also need to be further mentioned and analyzed in the discussion section. The discussion is still bland. For instance, the statement: "However, there was an emphasis additionally placed on safeguarding required to mitigate against over-measurement and the possible anxiety that elevated readings could create, though this was balanced with points made about elevated readings being positively viewed in terms of being a motivating factor to engage in lifestyle behaviours such as exercise to reduce BP.". How does the sentence support the authors' arguments in the discussion: "When compared to the distribution of current PPI events, [14] it is unclear if participation levels are representative of patients affected by each condition or treatment studied, or if the distribution of known and published PPI events match up to patient priorities more generally. The former in the results indicates a positive tone of the PPI initiative. However, the latter shows otherwise unless the authors provide thorough information to support their argument about the gap.

Reviewer #2: The report has some intrinsic limitations but the Authors have, nonetheless, acknowledged most of them at the end of the manuscript and have re-structured the paper according to recommendations raised during the Reviewing process.

I have no further comments.

7. PLOS authors have the option to publish the peer review history of their article (what does this mean?). If published, this will include your full peer review and any attached files.

Reviewer #1: No

Reviewer #2: No

---

## [Editor Report · Acceptance letter]

1 Dec 2022

PONE-D-22-13809R1 

Digital technology and patient and public involvement (PPI) in routine care and clinical research - a pilot study 

Dear Dr. Chen:

I'm pleased to inform you that your manuscript has been deemed suitable for publication in PLOS ONE. Congratulations! Your manuscript is now with our production department. 

Kind regards, 

on behalf of

Dr. Giuseppe Limongelli 

Academic Editor

PLOS ONE